# Hybrid Fruits for Improving Health—A Comprehensive Review

**DOI:** 10.3390/foods13020219

**Published:** 2024-01-10

**Authors:** Marta A. A. S. Cruz, Pedro P. S. Coimbra, Carlos F. Araújo-Lima, Otniel Freitas-Silva, Anderson J. Teodoro

**Affiliations:** 1Food and Nutrition Program, Functional Foods Laboratory, Federal University of the State of Rio de Janeiro, Rio de Janeiro 22290-240, RJ, Brazil; martasousacruz@gmail.com; 2Laboratory of Environmental Mutagenesis, Federal University of the State of Rio de Janeiro, Rio de Janeiro 22290-240, RJ, Brazil; pedro.coimbra@edu.unirio.br (P.P.S.C.); araujo.lima@unirio.br (C.F.A.-L.); 3Laboratory of Pharmaceutical and Technological Innovation, Department of Genetics and Molecular Biology, Federal University of the State of Rio de Janeiro, Rio de Janeiro 22290-240, RJ, Brazil; 4EMBRAPA Agroindústria de Alimentos, Rio de Janeiro 23020-470, RJ, Brazil; otniel.freitas@embrapa.br; 5Integrated Food and Nutrition Center, Department of Nutrition and Dietetics, Fluminense Federal University, Niterói 24020-140, RJ, Brazil

**Keywords:** hybrid fruits, compounds, bioactive, antioxidants, health

## Abstract

Several species of hybrid fruits, such as citrus, grapes, blueberries, apples, tomatoes, and lingonberries among others, have attracted scientific attention in recent years, especially due to their reported antioxidant and anti-inflammatory properties. The bagasse, leaves, bark, and seeds of these hybrid fruits have large amounts of polyphenols, such as flavonoids, which act as potent antioxidants. Several studies have been carried out in cellular models of neurotoxicity of the extract of these fruits, to document the beneficial effects for human health, as well as to prove its antiproliferative effect in cancer cells. In the present review, through a synthesis of existing information in the scientific literature, we demonstrate that hybrid fruits are a source of antioxidant and bioactive compounds, which act in the inhibition of diseases such as cancer, diabetes, and inflammatory and neurodegenerative diseases, and consequently improving human health.

## 1. Introduction

Brazil is the world’s third largest producer of fruits, behind China and India, with about 45 million tons per year. Most of this production is aimed at the domestic consumer market—only 2.5% is exported [1,2].

In 2017, the world commercial fruit production in general, according to FAO data, was about 865 million tons in an area of approximately 65 million hectares. China stands out with about 30% of all world fruit production and 24% of the world’s cultivated area for commercial purposes [1].

From north to south, Brazil has more than 2.5 million cultivated hectares. The production estimate reaches 33 million reais in gross value and the sector holds about 16% of the Brazilian agribusiness workforce, with millions of jobs generated. However, regarding exports, the country occupies the 23rd position, as it exported only 3% of what was produced in 2022 [3].

Fruit growing participates in various ways in the growth of the Brazilian economy, either as a source of food, bringing benefits to the population, or generating many jobs directly and indirectly. Even with a small performance in the international market, Brazil had an increase in the generation of foreign exchange in recent years, already with exports, both of fresh fruits and concentrated juices, indicating an increase in this international performance [2].

Brazil boasts the planet’s most extensive biodiversity, harboring over 15% of the global species count. This richness extends to its biota variety, encompassing more than 30,000 distinct species of angiosperms (flowering and fruit-bearing plants) distributed across the country [4].

Fruits are indispensable foods for proper body function due to their nutritional composition including vitamins, carbohydrates, proteins, fibers, minerals, and water [5].

The scientific evidence is based on the performance of fruits’ biochemical components in the prevention of cardiovascular diseases and various types of cancer, as demonstrated in scientific literature [5].

The rising prevalence of low-nutrient and monotonous diets contributes to the emergence of issues related to nutritional deficiencies and chronic diseases, contributing to malnutrition and imbalanced dietary patterns. This underscores the need for advancements in understanding the composition and health-promoting properties of overlooked and underutilized Brazilian native species [6].

The recognition of the nutritional value of tropical fruits has aroused interest in the scientific community, leading to an increase in their consumption locally and internationally. In order to understand, guarantee, and validate the nutritional potential, it is necessary to characterize the physicochemical properties of the fruits and the quantification of their bioactive compounds through clinical trials, scientific validation, transparent labeling, safety assessment, and quality control to ensure safety and efficacy while safeguarding consumers against misleading claims [6,7].

Different clinical and epidemiological studies have shown results associated with the consumption of fruits and vegetables, their multiple health benefits, and the decreased risk of cardiovascular diseases, diabetes, macular degeneration, age-related cataracts, and some cancers [8]. The benefits occur due to nutrients such as dietary fiber and bioactive compounds, vitamins A, B, C, and E, polyphenols such as flavonoids, tocotrienols, alkaloids, saponins, terpenoids, phytosterols, organosulfur compounds, lactones sesquiterpenes, carotenoids, thiocyanate, and selenium [9,10].

The human consumption of wild species of fruits is an ancient habit. Industrialization, interbreeding, and natural changes have played a key role in the evolution and vegetative propagation, since they allowed morpho-functional variations in leaves, flowers, and berries; they have also improved the quality and increased the number of existing cultivars by phenotypic plasticity [11].

From the perspective of fruit growers and producers, these novel fruits should possess distinct functional characteristics, including increased yields, enhanced resilience to climatic factors, and seedlessness. Additionally, to captivate consumers, they should boast improved aroma and palatability [12]. 

Natural antioxidants, despite their consumption still being limited, are of interest in the cosmetic, pharmaceutical, and food industries, to be used as substitutes for synthetic antioxidants and protect against the oxidative degradation of free radicals. Low levels of production, the lack of investment, or little knowledge of production or conservation systems leads to the limited use of some tropical fruits for direct consumption or the production of processed foods [6].

The improvement of perennial fruit crops depends largely on conventional methods of introduction, selection, or hybridization using the cultivated genotypes of a species. However, in most crops, most cultivars are developed with relatively narrow genetic diversity. It is estimated that 75% of the genetic diversity of crops has been lost in the twentieth century [13]. When species do not present the desired characteristics, their creators achieve these through methods such as mutation, polyploidization, or recombinant DNA technologies. The adaptation of the new species will be influenced by agriculture, due to the challenges imposed by climate change on wild plant forms. These are identified as critical components for food security and environmental sustainability [14]. To ensure and validate hybrid fruits for health, and for them to be accepted by consumers, they must be checked by regulatory authorities around the world, which guarantee against misleading claims. Thus, being recognized as safe (GRAS) or approved by the FDA would reinforce the credibility of their integration into diets [7].

Facilitating the transfer of genes between species enables the exchange of genomes, leading to alterations in the genotypes and phenotypes of offspring. Hybridization takes place in both prezygotic and postzygotic barriers within extensive barriers. Techniques such as chromosome doubling, species bridging, protoplast/somatic fusion, and embryo rescue play a crucial role in overcoming diverse barriers during wide crossings, ultimately aiding in the recovery of fertile progenies [14].

Considering this, hybridization involving wild relatives and related taxa has been increasing every day in recent fruit improvement programs. Alternative technologies, such as CIS genesis and genome editing, can facilitate the development of genetically modified crop varieties with multiple favorable traits. Through crop improvement, introducing beneficial foreign gene(s) or silencing the expression of the endogenous gene(s) in cropping plants, genetic engineering, and plant transformation have played a key role in improving crops [15].

## 2. Hybridization

### 2.1. Concept

Hybridization has an important role in the evolution of many lineages with great phenotypic diversity, because of the great availability of genomic tools as well as innovations in genomic analysis, as it causes a divergent flow between taxa, leading to an adaptation of these lineages to new environments, and contributing to hybrid speciation [16].

Since the beginning of human organization into societies, the improvement of livestock, farming, and food supplying performance has been a major concern. In search of a faster and more accurate way to modify food, scientists in the 20th century found a way to modify foods by altering their DNA through genetic engineering, turning these foods into genetically modified organisms (GMOs). Figure 1 shows the dates of development of genetically modified organisms which serve as food supply. In the 1940s, plant breeders used radiation or chemicals to randomly alter the DNA of an organism, basing in mendelian genetics knowledge. In 1953, based on Rosalind Franklin’s findings, scientists James Watson and Francis Crick purposed the double-helix model of DNA structure [17].

In 1973, biochemists Herbert Boyer and Stanley Cohen pioneered genetic engineering by transferring DNA from one bacterium to another. The Food and Drug Administration (FDA) granted approval for the first genetically modified organism (GMO), Human Insulin Product, in 1982, to treat diabetes. Following the Asilomar Conference on recombinant DNA and mutagenesis in 1975, the United States established federal regulations for biotechnology in 1986. This policy, jointly overseen by the FDA, USDA, and EPA, outlines safety regulations for GMOs. Consumer access to genetically engineered products began in 1990, with items like summer squash, soybeans, cotton, corn, papaya, tomatoes, potatoes, and canola. However, not all these products were available for purchase. In 1992, the FDA declared that safety standards for food from GMO plants must align with those for traditionally bred plants. The first GMO product, a genetically modified tomato, entered the market in 1994 after federal agencies confirmed its safety through evaluations comparable to traditionally grown tomatoes. In 2001, Brazil enacted laws mandating the labeling of packaged foods intended for human consumption that contained or were produced with GMOs. The criteria for existing limits underwent modifications in 2008 and were officially approved in 2015. By 2003, the World Health Organization (WHO) and the Food and Agriculture Organization of the United Nations (FAO) had established international guidelines and standards for determining the safety of genetically modified foods. In 2005, genetically modified alfalfa and beets became commercially available in the USA. Due to the susceptibility of citrus species, many were domesticated from wild ancestors or crossbred. In 2016, the U.S. Congress passed a law requiring labeling for certain food produced through genetic engineering, using the term “bioengineer”. GMO apples entered the market in 2017, and in 2019, the FDA concluded a consultation on the first food from a genome-edited plant. In 2020, U.S. consumers gained access to GMO pink pineapple [17,18,19,20].

Plant breeding is common across species, with the aim of, for example, increasing nutrient concentrations and some unique secondary metabolites; prolonging the shelf life of cut fruits, vegetables, and flowers; improving yield potential; and increasing tolerance to abiotic stresses and resistance to scourges and plant diseases [21].

Anthocyanins may be beneficial to human health as they are reported to inhibit certain cancers and degenerative diseases. In addition, anthocyanins are responsible for the coloration of flowers, fruits, and vegetables [22].

### 2.2. Hybridization Techniques

Over the past decade, green biotechnology has introduced a new generation of techniques, referred to as “new plant breeding techniques” (NPBTs), which it has been using as a powerful tool around the world. From many points of view, their potential is much greater compared to that of traditional reproduction and transgenesis (i.e., “classical” genetic engineering). The term NPBTs comprises several techniques, the best-known being “cisgenesis” and “genome editing through site-directed nucleases” [23]. These alternative technologies can address many of these issues and facilitate the development of genetically modified crop varieties with multiple favorable traits [24]. A set of alternative technologies have been used for the development of crop improvements in recent years, in response to public concern and lower consumer acceptance of transgenic crops. Figure 2 provides a summary of these techniques.

Genetically modified (GM) crops are those crop plants whose genome is modified using genetic engineering techniques to improve the existing traits or the introduction of a new trait that does not occur naturally in the given crop species [25].

The plants produced by the insertion of specific segments of foreign nucleic acid/gene sequence into their genome using transformation methods (such as *Agrobacterium*-mediated transformation or direct gene transfer) are known as transgenic plants. The inserted gene, also known as a transgene, may come from an unrelated plant, bacterium, virus, fungus, or animal species. Thus, the advent of genetic transformation overcomes the major limitation of conventional plant breeding in which sexual compatibility between species is a precondition to cross them [26,27].

Such techniques produce cultivation plants genetically like developed plants through reproduction. Moreover, they can be used to develop improved crop plants. This occurs because they project the final products (genome of the modified culture) so that they do not contain any stranger’s gene (transgene) [27].

Transgenesis and CIS genesis use the same molecular processes and techniques to transfer gene(s) to a plant, but the CIS genic plant will retain only species-specific genes that could also have been transferred by traditional breeding. Genome editing is responsible for the insertions or deletions of nucleotides at target sites that can cause genetic mutations, resulting in the silencing of the unwanted gene [15,28]. Among the genome editing technologies, the most employed ones are Zinc Finger Nuclease (ZFN), Activating Transcriptional Effector Nuclease (TALEN), and Clustered Regularly Interspaced Short Palindromic Repeats (CRISPR), in order of appearance. The CRISPR/Cas9 system is currently the most current tool for genome editing, due to its simple structure and applicability to a wide range of species [26,29].

The 21st century is recognized as the post-genomic era, marked by the accessibility of genomic sequence data for diverse crop plants, transforming plant-breeding initiatives. The adoption of advanced synthetic biology tools, referred to as “genome editing tools,” seamlessly incorporates desired traits into crop genomes. Synthetic biology employs the methodical design of biological molecules to attain specific objectives. These tools, guided by the systematic design principles of synthetic biology, operate with precision, accuracy, and predictability [20].

### 2.3. Nutrition and Sensorial Aspects of Hybrid Fruits

The development of hybrid plants involves crossbreeding distinct varieties or species. Hybridization within a single species may result in a phenomenon called heterosis. Individuals exhibiting heterosis are distinguished by increased fertility, enhanced lifespan, and greater fruitfulness. The newly produced fruits, in addition to their functional and health properties, also showcase novel organoleptic attributes [30].

The natural or intentional generation of interspecific hybrid fruits leads to the development of new secondary metabolites and heightened activity in existing biosynthesis pathways. Studies indicate that these hybrids may exhibit improved characteristics inherited from both parental species, particularly concerning the total content of phenolic compounds and their distribution in individual plant organs [31,32,33].

Studies conducted to date have proven that the phytochemical and organoleptic characteristics of interspecific hybrids may be better compared to parental species, whose fruits generally have an increased content of simple sugars. In analyses with the hybrids of pummelo and Oro Blanco, a hybrid of pummelo, the latter was demonstrated to have a sweeter flavor, with smaller amounts of simple sugars, being healthier for diabetics, so that the hybridization process not only enhanced the taste properties but also led to a reduction in the sugar content of the Oro Blanco fruit when compared to its parent fruit, the pummelo [34,35,36,37,38,39,40].

The findings from physicochemical and biochemical studies unveiled that substances such as stilbenes have health-promoting biological effects related to the prevention of coronary heart disease and the reduction of cancer, aligned with the ability to inhibit the enzymatic activity of the nuclear transcription factor NFκB crucial transcriptional mediator in cell activation pathways, involved in the signaling of neoplastic, inflammatory, and degenerative diseases [40].

Studies with hybrid citrus fruits (*Citrus aurantium*) have demonstrated one hundred and two chemical constituents that have been identified from their pulp and peel, including volatile oils, terpenoids, phenols, limonin, sugars, etc. [41,42].

The production of hybrid fruits increases day after day, as well as their storage technologies. Their medicinal applications have attracted the attention of the scientific community, due to their various known pharmacological effects, such as anti-inflammatory, antioxidant, antitumoral, hypolipidemic, and chemoprotective effects on organs [43].

Tropical fruits and subtropical fruits are recognized as a source of a high content of bioactive compounds and health-promoting properties due to their nutritional composition. These beneficial health effects are related to the content of several of these bioactive compounds, primarily flavonoids, and non-flavonoid phenolics. Many of these compounds are common in different tropical fruits, such as epicatechin in mango, pineapple, and banana, or catechin in pineapple, cocoa, and avocado [44,45].

The hybrid fruit (*Aronia prunifolia*) of aronia, also called chokeberry, was found to have more antioxidants than the non-hybrid fruit [46].

Vitis grapes are considered rich sources of phenolic compounds belonging to groups such as anthocyanins, flavanols, tannins, phenolic acids, and stilbenes [47].

Research indicates a correlation between the intake of V. vinifera fruits and a decrease in the incidence of cancer, through the inhibition of NFκB, in the early stages of the development of, e.g., skin cancer, and malignant melanoma [48,49].

Studies on the molecular mechanisms of antioxidant substances present in hybrid fruits that act as anti-inflammatories, point to the modulation of the activity of various proteins and transcription factors, which are involved in metabolic pathways related to the synthesis of cytokines, chemokines, and adhesion molecules [50].

However, it should be noted that the phenolic content varies between qualitative and quantitative intrinsic (genus, species, and cultivar) and extrinsic (environment, cultivation, handling, and storage) factors, in addition to depending on factors such as extraction and quantification methods. Depending on each region, and its particularities, fruits will be differentiated in terms of chemical composition, consequently having distinct biological activities, with the study of cultivars planted regionally proving to be of fundamental importance [51,52].

## 3. Antioxidants

The emergence and heightened activity of novel secondary metabolites within current biosynthesis pathways result from the phenomenon of natural or planned generation of interspecific hybrid fruits Studies have shown that hybrid fruits, both in the total content of phenolic compounds and in their distribution, may exhibit improved traits inherited from both parental species [53,54].

Scientific research has demonstrated that the phytochemical and organoleptic attributes of interspecific hybrids may surpass those of parental species, whose fruits typically feature elevated levels of simple sugars [12,34].

The results of physicochemical and biochemical studies revealed that substances such as stilbenes, normally abundant in *Vitaceae* spp., have health-promoting biological effects related to the prevention of coronary heart disease [32], and the reduction of the occurrence of cancer [36,54,55].

Flavonoids make up a large class of secondary metabolites of low molecular weight that are present in almost every compartment of plants, from the roots to the flowers and fruits [56]. Flavonoids, a category of polyphenolic compounds, are classified into six groups: iso flavonoids, flavanones, flavanonols, flavanols, flavones, and anthocyanidins, and are present in a diverse range of plants [57].

Flavonoids possess the ability to inhibit auto-oxidation and scavenge free radicals. Functioning as antioxidants, they can transfer electrons to free radicals and serve as catalysts for chelated metals [58].

In the presence of biotic and abiotic stressors, such as drought, wounds, and metal toxicity, numerous genes involved in flavonoid biosynthesis are activated, leading to an elevation in flavonoid levels [59,60].

As can be seen in Figure 1, extracts of hybrid fruits rich in polyphenols come into contact and cause a reduction in cancer cells, which, in addition to natural internal factors, have gone through external factors and have their ROS production increased [61].

Anthocyanins belong to the flavonoid class; they are highly hydrophilic and are responsible for the red and blue colors in fruits, vegetables, and flowers [33]. In addition to being natural pigments, they are potent antioxidants and can prevent lipid oxidation and eliminate free radicals [54]. It has been reported that the dietary intake of fruits and vegetables has beneficial effects on human health, such as anticancer and anti-aging properties [62,63] Anthocyanins are also important for improving the nutritional values of processed foods [64].

Anthocyanins are being used as one of the most promising ingredients in the food, beverage, cosmetic, and nutraceutical industries. The nutraceutical activities as well as the anticancer activities of anthocyanins and anthocyanidins have been extensively reviewed [64].

The abilities of anthocyanins to induce apoptosis and suppress angiogenesis have been explained as the reasons for the anticancer activities of anthocyanins, which cause cyanidin-3-glycoside (C3G) to block the ethanol-induced activation of the ErbB2/cSrc/FAK pathway, preventing cell migration/invasion. This effect is beneficial for preventing ethanol-induced breast cancer metastases [40,49,62].

There is a great demand for natural antioxidants, arising from the desire to switch from synthetic to natural products, which drives research in this area [62]. The antioxidant capacities of polyphenols in general and anthocyanins in particular are evaluated by various methods. TEAC (Trolox Equivalent Antioxidant Capacity), FRAP (Ferric Reducing Ability of Plasma), and ORAC (Oxygen Radical Absorbance Capacity) are just a select few from the numerous trials commonly employed in the scientific community.

Many members of the anthocyanin family have demonstrated antioxidant activities like α-tocopherol, Trolox, quercetin, and catechin. Table 1 illustrates the experimental antioxidant capacities of hybrid fruits, together with their total polyphenolic content, calculated as gallic acid equivalents. Current scientific research has considered very carefully the content of fruits, especially in sugar, vitamins, and minerals, so that they have better quality and meet the requirements of consumer needs. The hybridization techniques have produced hybrid fruits with a higher content of mineral elements and good quality compared to the native fruits, according to Table 2, which includes data collected in studies carried out on hybrid and native fruits.

The firmness of the fresh fruit and the sugar content are important quality factors, as they directly influence the purchasing power of consumers. The taste is the most complex to analyze since the organoleptic quality of the fresh fruit is composed of various organic acids, along with soluble sugars and aromas. Eleven organic acids were identified in the apple pulp and five more in the whole fruit, with malic acid, the predominant organic acid in apple fruits, directly influencing the flavor. The concentration of ascorbic acid in the fruit progressively decreases from the peel to the core of the fruit. Vitamin C is present in two forms: in ascorbic acid and in its oxidized form, dehydroascorbic acid [65,66,67].

Vitamin C participates in the processes of regulating cell growth, cell signaling, apoptosis, antioxidants, and as a cofactor for enzymes. Vitamin C comes mainly from the consumption of fruits and vegetables, being reduced by heat during processing; therefore, its nutritional value in raw foods is higher than in processed form. Vitamin C eliminates reactive oxygen species (ROS) and reactive nitrogen species (RNS) and regenerates α-tocopherol and coenzyme Q from α-tocopherol and coenzyme Q radical α. Studies postulated that ascorbate induces the decomposition of lipid hydroperoxide into genotoxins in the absence of redox-active metal ions, leading to a reduction in the growth of aggressive tumor xenografts [9,30,31]. In tests performed in animal species, the consumption of fruits rich in vitamin C helped protect the body against cardiovascular disorders, gastrointestinal disorders, cancer, skin infections, and diabetes through reduced insulin glycation and an increase in glucose homeostasis [31].

Sugars, acids, and aromatic compounds are recognized as major components in fruit quality. These components vary greatly among the different varieties of fruits [46].

A comparative table with data from several studies is presented, with the values of antioxidant capacity, vitamin C determined, total polyphenol amounts, percentage of sugar, and water of the native cultivars, because they are closely related to the quality of hybrid fruits produced.

The results of this table show that the nutritional values of the fruits can be altered by the application of hybridization, making them a qualitatively different product.

## 4. Health Benefits of Hybrid Fruits

The increased antioxidant capacity in the fruit increases its action against some diseases that compromise human health, as demonstrated during this study. Therefore, since hybridization increases the number of antioxidant compounds, as well as the antioxidant capacity of the fruit in the human body, we will describe some beneficial effects that the consumption of hybrid fruits can bring to those who use them in their diet [69].

In all countries, regardless of whether they are developed or developing, there is an increase in the interest in “natural” products, and their content of phytochemicals and antioxidants, for the treatment of human diseases. Some of these natural products rich in antioxidants include tea, shrubs, wine, and fruit juice; when produced with the hybrid fruits, they will be much richer, which may encourage the increased production of these highly oxidizing fruits [70,71].

The preferential use of natural products occurs mostly for their minimal side effects and the growing preference for natural products used in preventive and therapeutic medicine. There is a great tendency to use natural vitamins C and E as defense and protection against ultraviolet radiation in skin and other cosmetic products [72].

Fruit juice, wine, and processed food drinks are currently supplemented with fruit-derived ascorbic acid. Natural dyes derived from carotenoids obtained from antioxidant-rich fruits and vegetables are used in the replacement of synthetic artificial colors in the manufacture of food products.

In vitro health research on antioxidants contained in hybrid fruits, strengthened by existing studies on native fruits, is still being conducted in experimental models, with many of the claims about their therapeutic effects yet to be verified and confirmed, especially in humans. Research into phenolic compounds is of increasing interest due to the vital biological and pharmacological characteristics that these antioxidants have demonstrated in human health [72]. Seeing “bioactive” as the topic of discussion in most congresses related to food and health highlights the emphasis on the functional properties of these food sources. This reinforces the need for more research to verify and substantiate these health claims and to determine the antioxidant contents of fresh peeled hybrid fruits versus “in natura” (consumed with peel) or processed (dried), as normally incorporated in other food products.

### 4.1. Anticancer Effect

Cancer is a worldwide health concern. This disease is caused by irregular cell growth with invasive potentials. The discovery of new bioactive components can be considered a new therapeutic strategy for cancer protocols [73,74,75].

During the past two decades, plant studies have found bioactive compounds that have been reported as new health agents for the prevention and/or mitigation of different human diseases such as cancer and inflammation, cardiovascular, and neurodegenerative diseases [76].

The generation of free radicals (ROS) is unavoidable during various physiological reactions in the human body, including respiration, as the mitochondrial electron transport chain directly reduces oxygen using the free energy of electrons. Additionally, free radicals are produced as a natural outcome of metabolic processes and due to external influences (Figure 3), such as exposure to smoke, pollution, chemicals, ozone, ultraviolet radiation, X-rays, pesticides, and specific drugs [77,78].

Nitric oxide (NO) functions as a signaling molecule with significant roles in both physiological processes and cancer promotion. It is proposed that low levels of NO contribute to cancer promotion, whereas elevated levels of NO serve as protective factors against cancer. Both reactive oxygen species (ROS) and various reactive nitrogen species (RNS) possess carcinogenic potential by modulating the inflammatory state and influencing cellular lipid structures, angiogenesis, anti-apoptotic pathways, and more. For instance, low concentrations of NO can induce redox imbalance, heightened inflammation, and damage to subcellular components, thereby accelerating the neoplastic process [78].

In the promotion of the tumor, paraneoplastic cells in active proliferation accumulate in a relatively long and reversible process. Progression, the final stage of neoplastic transformation, involves the growth of a tumor with invasive and metastatic potential [75].

Research already carried out shows the presence of important antioxidant and antitumor activity in the different hybrid cultivars cultivated in the Brazilian territory.

Numerous studies have demonstrated that flavonoids exhibit the capability to neutralize free radicals, regulate cellular metabolism, and prevent diseases associated with oxidative stress [78].

Cancer is a diverse condition marked by unregulated cell proliferation and disrupted cell cycle, resulting in the development of abnormal cells that invade and metastasize to other regions of the body [78,79]. 

Cancer stems from internal factors such as oxidative stress, hypoxia, genetic mutations, and deficient apoptotic function, while external factors include heightened exposure to stress, pollution, smoking, radiation, and ultraviolet rays [59]. Key characteristics of cancer cells encompass altered metabolism, disrupted cell cycles, frequent mutations, resistance to immune response, chronic inflammation, metastasis, and the induction of angiogenesis [80]. Numerous studies establish a connection between cancer and metabolic disease, influenced by diverse degrees of mitochondrial dysfunctions and metabolic alterations [80,81].

Mitochondria play a crucial role in providing cellular energy, regulating metabolism, signaling cell death, and generating reactive oxygen species (ROS). Tumor cells undergo significant metabolic alterations, including heightened aerobic glycolysis, dysregulated pH, impaired lipid metabolism, increased ROS production, and compromised enzyme activities [61] (Figure 3). Conversely, the extracellular environment becomes more acidic, favoring inflammation, leading to increased glutamine-driven lipid biosynthesis and upregulating pathways associated with tumorigenesis and metastasis. Additionally, cardiolipin levels decrease in membranes, impairing enzyme activities, mitochondria become hyperpolarized, and this effect correlates with the malignancy and invasiveness of cancer cells [81].

A study with luteolin, a flavonoid found in different plants, demonstrates that the flavonoid acts as an anticancer agent against various types of human malignancies, such as lung, breast, glioblastoma, prostate, colon, and pancreatic cancer. It also blocks the development of cancer in vitro and in vivo by inhibiting the proliferation of tumor cells, protecting against carcinogenic stimuli and the activation of cell cycle arrest, and inducing apoptosis through different signaling pathways. This can also reverse the transition of epithelial-mesenchymal cancer cells through a mechanism that involves shrinking the cytoskeleton, inducing the expression of the epithelial biomarker E-cadherin and the negative regulation of mesenchymal biomarkers -cadherin, and vimentin [39,48] (Figure 3).

Flavonoids exhibit various anticancer effects, including the modulation of enzymatic ROS-killing activities, involvement in cell cycle disruption, the induction of apoptosis and autophagy, and the suppression of the proliferation and invasion of cancer cells [40].

There is scientific evidence that there is a need for a balance between oxidants and antioxidants to maintain health, as changes in this balance can lead to pathological responses that result in functional disorders and diseases. Studies also suggest that a diet excluding vegetables and fruits may alter hormone production, metabolism, or action at the cellular level of the individual, thereby increasing the incidence of breast, colorectal, and prostate cancer [78,79,81].

#### Cytotoxic Activity

For studies of carcinogenesis and mutagenesis, cell lines such as MCF-7 (breast), Hep-2 (larynx), PC-3 (prostate), DU-145 (prostate), HeLa (cervix), HT-29 (colon), and OVCAR03 (ovarian) are often used, which are specifically derived from the transformation of cells of a type of carcinoma [79]. Most of these cell models have similar characteristics with rapid proliferation and a cell cycle of 18 to 24 h, which may even influence the verification of similar cytotoxic activities between the various cell lines [49]. The drugs tested for possible cytotoxic activities in the face of this cellular model are kept in contact with the cells for a period of 24 to 72 h, determining the dose–effect curve and subsequently the proposed inhibition parameter [76].

Cell death can be classified according to characteristics that are morphological, apoptosis-, necrotic, autophagic, or mitosis-associated [79,80].

In necrosis, an “accidental” cell death occurs, as it is usually the result of an unintentional traumatic injury, which may be thermal, chemical, or due to lack of oxygen, where the cell enlarges its volume, causing cellular disruptions, randomly releasing fragments in its surroundings [81].

Autophagy is an evolutionarily conserved and genetically controlled adaptive process, which occurs in response to metabolic stress resulting from the degradation of cellular components [82].

Currently, the cytotoxic in vitro activity of tumor cell cultures becomes important for the identification of anticancer agents. Cell cytotoxicity has been evaluated by several methods. Among the methods to evaluate viability, the MTT (3-(4,5-dimethylazol-2-yl)-2,5-diphenyltetrazolium bromide) assay stands out because it is an indirect, accurate, and rapid test based on a colorimetric reaction. The MTT salt of yellow color, when incubated with metabolically active cells, enters the mitochondria and is reduced by the enzyme succinate dehydrogenase, producing crystals of formazan, with dark blue coloration; thus, the resulting optical density is determined in a spectrophotometer. There are alternatives to MTT, such as MTS (salt of 3-(4.5-dimethylthiazole-2-yl)-5-(3-carboxymethoxypenyl)-2-(4-sulfophenyl)-2H-tetrazoli) which presents basically the same principle, but with lower toxicity and higher solubility in water [35].

*Aronia melanocarpa*, *Aronia arbutifolia*, and *Aronia prunifolia* demonstrated gradual inhibition in tumor cell proliferation after a 48-h treatment, proving to be cytotoxic to HeLa, HepG2, and HT-29 cells [37,46].

Four hybrids of *Malus sieversii* (red pulp apple) were compared with Fuji apple and showed a higher amount of phenolic and flavonoid compounds, with the hybrids “A38” and “Meihong” having a higher antioxidant and antiproliferative activity in human breast cancer in the strains MCF-7 and MDA-MB-231 [83].

Regarding flavonoids, in the extract of the leaf of atemoya (*Annona atemoya*), a hybrid of pinecone (*Annona squamosa*), and cherimoya (*Annona cherimola*), the metabolite rutin was identified in the regulation of the inhibitory effect in Alzheimer’s disease by avoiding β-amyloid aggregation and neuronal cell death [84].

Grapefruit, a citrus fruit resulting from the crossing of pomelo with orange, has as a secondary metabolite nootkatone, which, when tested in lung adenocarcinoma, was found to inhibit cancer progression via cyclooxygenase (COX-2), and consequently to inhibit the growth and decrease cell proliferation, by activating AMPK (AMP-activated protein kinase) [79].

Studies addressed in this review indicated the potential anticancer effect of antioxidant substances present in hybrid fruits, and in particular their capacity to inhibit the proliferation of cancer cells due to their effects on the cell cycle.

To be considered antiproliferative, food must have a rate of cell growth inhibition greater than 50% [40].

### 4.2. Anti-Diabetic Effect

Diabetes mellitus stands as a prominent contributor to both mortality and morbidity. Recent data indicate that approximately 150 million individuals worldwide are affected by diabetes, and this figure is projected to potentially double by the year 2025. Improvement in diabetes mellitus, digestive problems, immune disorders, cataracts, bronchitis, asthma, and other respiratory syndromes have been reported after regular intake of antioxidants from fruits and vegetables [36].

Like other chronic illnesses, diabetes poses a financial burden on individuals in terms of personal income, as well as on community productivity, given the escalating and substantial demands placed on healthcare and rehabilitation facilities. This results in direct and indirect economic costs associated with diabetes, placing a strain on the nation’s healthcare expenditure [64].

Research on the liver antioxidant defense in diabetic mice involving bitter orange extracts has revealed a substantial reduction in the blood glucose levels of the experimental diabetic mice compared to their untreated counterparts. In the liver of diabetic mice, there was an elevation in the activities of superoxide dismutase, while the activities of glutathione peroxidase, malondialdehyde, and nitric oxide were significantly decreased [42]. The hybrid orange extract exhibited the ability to enhance the liver’s antioxidant activity and contributed to the reduction of liver damage, as evident in the histological analyses of the experimental diabetic mice in comparison with untreated diabetic mice [85].

Intensive progress is underway in additional molecular and genomics endeavors aimed at enhancing nutritional value and other desirable traits. This endeavor is expected to broaden the applications of species and hybrids within the food and biotechnology industries [86].

### 4.3. Anti-Inflammatory Effects

Inflammation causes edema, resulting from the increased permeability of the endothelium tissue and the invasion of blood cells into the resulting space in response to an external stimulus. Macrophages, neutrophils, and epithelial cells usually mediate inflammation. The main pro-inflammatory mediators are prostaglandinsE2 and nitric oxide, which promote inflammation. Studies identify hybrid citrus species with various inhibitory metabolites of pro-inflammatory mediators, and report that flavonoids, coumarins, and essential oils found in some citrus species are effective against inflammation and have the potential to protect against inflammatory diseases [69,75].

Research was carried out on the methanolic extract of the leaves of cajazeira (*Spondias mombin*), a hybrid fruit rich in carotenoids and vitamin A, containing β-cryptoxanthin as its main carotenoid, followed by lutein. In phytochemical studies, its leaves showed the presence of tannins, saponins, resins, sterols and triterpenes, flavonoids, and alkaloids, demonstrating antibacterial activity against *Pseudomonas aeruginosa* and *Shigella dysenteries*, while extracts from the bark of the stem inhibited the growth of the bacteria *Escherichia coli* and *Klebsiella pneumoniae* [87,88].

Acute toxicity studies were performed in rats, with the induction of acute pancreatitis (AP) and with a diet composed of Granny Smith apple, which is a hybrid variety, reported as having the second highest number of flavonoids and procyanidins among all varieties of apples. The results of the studies indicated a protective effect on experimentally induced AP in rats by L-arginine. GSAE (Granny Smith Apple Extract) administration exhibited beneficial effects by reducing oxidative and nitrosative stress and modulating the inflammatory process [89,90].

### 4.4. Anti-Degenerative Disease Effect

New epidemiological studies have proposed an inverse correlation between high antioxidant intake from fruits and vegetables and many degenerative diseases and aging.

Neurodegenerative diseases such as Alzheimer’s disease (AD), Huntington’s disease (HD), Parkinson’s disease (PD), scrape induction disease, and amyotrophic lateral sclerosis (ALS) are characterized by the abnormal accumulation of proteins, either intra neuronally or extra neuronally. For example, Alzheimer’s disease (AD) is caused by a decrease or deficiency in acetylcholine, found in the synapses of the cortex, over a period [50,69,91].

The prevalence of these diseases, along with the associated concerns, has been on the rise due to the aging of the population, rendering the topic highly pertinent. Various pathways, including apoptosis, autophagy, oxidative DNA damage, and repair, have been implicated in different neurodegenerative diseases [91]. Each disease presents distinct mechanisms that remain largely unclarified, posing a significant challenge in the quest for potential therapies aimed at delaying aging effects and preventing these conditions [92,93].

Fruit hybridization, which has been proven to increase the amount of antioxidant compounds, has become an area of interest as a natural preventive/therapeutic strategy, as these substances have the potential to safeguard neurons from oxidative stress, regulate cell signaling pathways, and inhibit neuroinflammation. Mitochondria play a role in the pathophysiology of inflammatory diseases, contributing to the targeted production of free radicals within the cell (cell signaling and inflammatory process) and the detoxication of these same radicals in other situations [94,95].

### 4.5. Drug-Food Synergy

During the prescription of medications, or the formulation of herbs, supplements, or drugs, importance is not given to the drug–food synergy. This neglected fact, during consumption, can lead to health problems for those who seek a cure for a certain disease. Studies with patients treated with fruit juice and the drug donepezil. Sridharan and Sivaramakrishnan’s research indicated that the extract from *Citrus aurantium* L., containing nobiletin, demonstrated a deceleration in the progression of Alzheimer’s disease. Furthermore, the consumption of orange juice did not impact the participants’ plasma exposure when treated with cyclosporine [69,93,95]. Additionally, poly methoxy flavones derived from Citrus sinensis waste residue exhibited a synergistic effect when combined with anethole. However, it was observed that citrus, especially psoralen-rich foods, could increase the risk factor for malignant melanoma [96,97]. Various citrus fruits were found to influence drug-metabolizing enzymes; for instance, grapefruit contains a compound inhibiting cytochrome P4503A4 (CYP3A4), a key enzyme in drug metabolism. Studies revealed that furanocoumarins in grape juice, while inhibiting bowel cancer CYP3A4, enhanced the oral bioavailability of drugs like felodipine and midazolam. This, in turn, could lead to elevated toxic concentrations, potentially resulting in fatal rhabdomyolysis [77,91,98].

Pharmaceutical drugs like donepezil and naturally occurring galantamine have proven to be effective therapeutic agents for Alzheimer’s disease (AD). Nevertheless, reports have highlighted their adverse side effects on the gastrointestinal tract [69,97].

## 5. Conclusions

This review demonstrates that research into different fruits in their hybrid forms has proven their increased antioxidant activity, demonstrating their functional and nutritional potential. When compared to their native forms, they demonstrate a more effective chemical and biochemical composition for health, reinforcing the idea that we need to continue to study and develop more optimized hybridization techniques to ensure safety for consumers by increasing the consumption of these fruits. Based on studies like this and with the many health claims attributed to the antioxidant content of hybrid fruits, their sustained and increased production will lead to the increased consumption of these products by humans and a decrease in degenerative diseases. Thus, we suggest more studies, mainly on humans, with hybrid fruit extracts, as we need more research and clinical studies for a more in-depth analysis in search of understanding the underlying mechanisms, in order to reduce the effect of diseases and side effects of medications.

## Figures and Tables

**Figure 1 foods-13-00219-f001:**
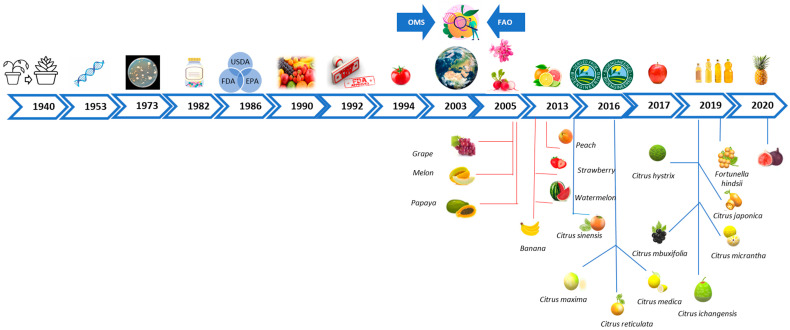
Timeline of genetic modification in agriculture, including some of the genomes of plants sequenced in the period indicated below, such as citrus; see related blue lines. In addition, other crops are indicated with red lines. Source: Authors, 2023.

**Figure 2 foods-13-00219-f002:**
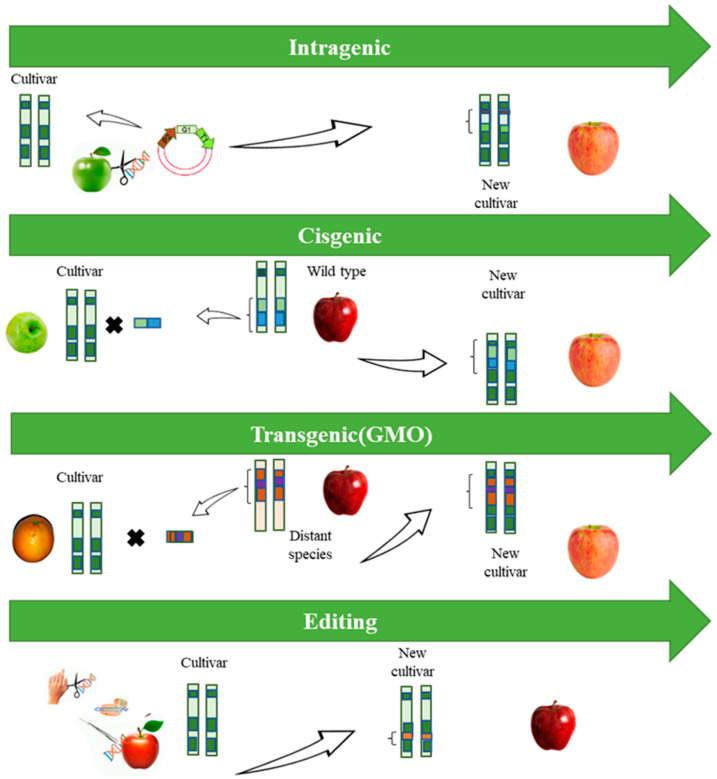
Schematic illustration of the different types of hybridization techniques. Source: Authors, 2023.

**Figure 3 foods-13-00219-f003:**
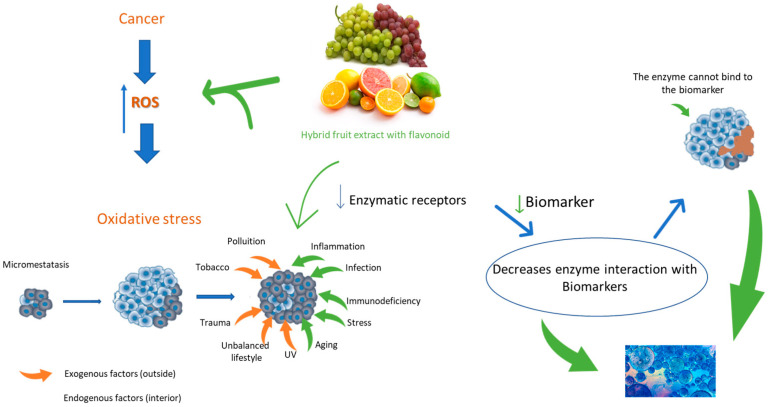
The flavonoids of hybrid fruits protect against ROS induced by metabolic alteration occurring in tumor cells.

**Table 1 foods-13-00219-t001:** Antioxidant capacity, total polyphenolic content, flavonoids, and anthocyanins from hybrid fruits.

Hybrid Fruits	TEAC (mmol Trolox/100 FW)	FRAP (mmol Fe^2+^/100 g FW)	ORAC (mmol Trolox/100 g FW)	DPPH (mmol TE/100 g DM)	Total Phenolics (mg GAE/100 g FW)	Total Flavonoids (mg/100 g)	Total Antocianins (mg/100 g)	References
melon	nd	nd	nd	nd	715.8	nd	nd	[13]
Grape (*Sweet jubilee*)	nd	63.7	nd	nd	2038	nd	nd	[14]
Mandarin	61	674,5	407.4	54.03	467	577	nd	[17]
Orange	nd	755	244	16.1	1341	nd	nd	[17]
Pummelo	24.9	7.31	449.1	10.1	1754	nd	nd	[17]
Grape (Vitis)	nd	42.3	58.5	33.7	1015.2	122	3457.5	[30]
chokeberry	nd	39.0	555.5	53.78	2340	556	256.4	[32]
SweetOrange	31.9	nd	nd	12.3	nd	nd	nd	[33]
Kumquat	15.7	nd	nd	2.9	nd	nd	nd	[33]
Lemon	21.2	nd	nd	8.8	nd	nd	nd	[33]
Strawberry	265.9	339	253.2	nd	6238	675	nd	[60]
Red Plum	185.3	208.2	274.9	nd	352	nd	nd	[60]
Apple	447	402	578	nd	490	nd	nd	[60]
Tomato	269	501	459	nd	310	nd	620	[60]

Values are expressed as the mean with standard deviation according to the literature. nd = not determined.

**Table 2 foods-13-00219-t002:** Chemical and biochemical components of hybrid fruits with their natives.

Fruit	Hybrid Fruit (H)	ORAC (mmol TE/100 g)	FRAP (µM Trolox/g FW)	DPPH (µM Trolox TE/g FW)	Total Polyphenols (mg GAE/100 g)	Sugar	Vitamin C (mg/100 g) Fresh Content	References
Native Fruit (N)	(%)
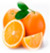	Sweet Orange (H)	468.96	nd	568.39	248.00	nd	49.90	[17,65,66,67]
Orange (N)	213.25	nd	333.76	206.00	nd	68.50
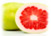	Pummelo (H)	319.01	nd	nd	275.50	nd	80.00	[17]
Pummelo (N)	201.56	nd	nd	172.00	nd	57.00
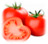	Tomato (H)	855.83	nd	nd	52.21	nd	30.40	[29,57]
Tomato (N)	406.27	nd	nd	28.18	nd	21.20
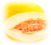	Melon (H)	nd	378.80	nd	96.00	3.97	18.34	[68]
Melon (N)	nd	493.80	nd	115.20	6.33	22.47
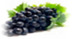	Grape (Vitis) (H)	341.00	nd	386.20	203.50	15.50	3.20	[14,30,69]
Grape (RSG) (N)	146.57	nd	169.45	367.00	16.67	10.90
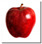	Apple (H)	nd	nd	72.20	588.90	12.34	9.80	[66]
Apple (N)	nd	nd	131.60	83.00	nd	8.81
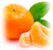	Mandarin (H)	357.54	nd	nd	211.10	nd	50.00	[17,67]
Mandarin (N)	163.88	nd	nd	195.80	nd	40.50

(H) = hybrid fruits; (N) = native fruits; RSG = red seedless grape; nd = not determined.

## Data Availability

The original contributions presented in the study are included in the article, further inquiries can be directed to the corresponding authors.

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
