# Peer review of "Hybrid Fruits for Improving Health—A Comprehensive Review"

_foods, 2024, doi:10.3390/foods13020219_

Round 1

Reviewer 1 Report

Comments and Suggestions for Authors

The integration of hybrid fruits into dietary practices has emerged as a compelling avenue in the pursuit of enhanced health and nutrition. This comprehensive review endeavors to consolidate and analyze the burgeoning body of knowledge surrounding the potential health benefits, nutritional value, and safety considerations associated with the consumption of hybrid fruits. By amalgamating insights from diverse scientific studies, this paper aims to elucidate the diverse spectrum of hybrid fruits, exploring their composition, bioactive components, and their purported contributions to holistic well-being. Furthermore, this review delves into the regulatory frameworks and safety evaluations governing these novel fruit hybrids, shedding light on their acceptance within the purview of governmental agencies and scientific communities. Through an extensive survey of literature and critical synthesis, this review endeavors to navigate the landscape of hybrid fruits, offering a nuanced understanding of their role in fostering improved health outcomes and dietary diversity.

The authors should try to improve the paper's quality by following the next suggestions:

Abstract:

The study does not offer a demonstration of findings as it is not a research paper. It effectively provides an overview and synthesis of existing information on hybrid fruits as an alternative for health improvement.

Introduction (Lines 61-64):

The mentioned approach seems to be aligned with the principles outlined in the provided paper (https://doi.org/10.3390/foods12214001). Incorporating similar methodologies could enhance the clarity and depth of this study's approach.

Introduction (Repetitive References):

The introductory section seems to have certain references reiterated, potentially elevating the similarity index of the paper. Diversifying the sources cited could enhance the comprehensiveness of the introductory part.

Line 100-104:

Consider revising this segment for greater clarity as the intended message might not be distinctly conveyed. A clearer articulation of the concept would enhance the reader's understanding.

Safety and Approval of New Fruits:

Have these newly introduced fruits been acknowledged as safe for consumption? Clarifying whether they are Generally Recognized as Safe (GRAS) or FDA-approved would bolster the credibility of their integration into diets.

Figure 1:

While the visual presentation in Figure 1 is commendable, the size renders it difficult to read. Enlarging the figure would significantly improve its accessibility and utility for readers.

FDA Mention (Line 140):

For the first instance, it's advisable to spell out the abbreviation "FDA" as "Food and Drug Administration" for clarity before using the acronym.

References in Section 2.3 and Lines 139-163:

Consider adding pertinent references to substantiate the claims made in these sections. Strengthening these assertions with reliable sources would fortify the paper's credibility.

Section 2.3:

The structure and clarity of Section 2.3 could benefit from improvement. Enhancing the organization and coherence of this section would enhance its readability and comprehension.

Tables 9 and 10:

Tables 9 and 10, currently presented as images, require better clarity and accessibility. Transforming them into readable formats and discussing the data within these tables would enhance their informative value.

Proper Citation (Line 629):

Ensure proper citation of references for accuracy and academic integrity.

Visuals:

The colorful and logical presentation of images is highly commendable. They effectively complement the content.

In conclusion, enhancing the logical sequencing of content, providing more comprehensive data, specifying the discussed hybrid fruits, and reinforcing claims with relevant references would substantially augment the quality and impact of this paper.

Author Response

Anderson Junger Teodoro, PhD

Department of Nutrition and Dietetics

Universidade Federal Fluminense

email: atteodoro@gmail.com

Editor

Food Journal

Foods(https://www.mdpi.com/journal/foods)

Dear Editor,

Find an electronic version of the revised manuscript entitled Hybrid fruits as an alternative for improving health - a comprehensive review. (Submission Ref. Manuscript ID: foods-2784565).

We would like to thank the reviewers for all their comments and suggestions that helped us improve our work and present it in a more acceptable way. The response to the comments raised by the reviewers, as well as the list of changes made to the revised manuscript, are attached below, in red.

All changes made to the manuscript have been highlighted as requested. We hope that this revised version complies with the Reviewers' suggestions and is suitable for publication in his journal.

Best regards,

Anderson Junger Teodoro, PhD

Highlights comments:

The version of the manuscript found in the link above was used for the Reviews.

https://susy.mdpi.com/user/manuscripts/resubmit/c76bd2da5afbba04ed9c4122ac558cdc

(I) All references are relevant to the content of the manuscript.

(II)  The revisions of the manuscript were highlighted, so that any of changes can be easily reviewed by editors and reviewers.

(III) This cover letter explains, point by point, the details revisions of the manuscript and their responses to the reviewers Comments.

(IV) All comments in the review are explained in the resource.

(V) The revised version is attached to the editors and reviewers.

Reviewer #1:

Quality of English Language

() I am not qualified to assess the quality of English in this paper
( ) English very difficult to understand/incomprehensible
( ) Extensive editing of English language required
( ) Moderate editing of English language required
( ) Minor editing of English language required
(x) English language fine. No issues detected

Comments and Suggestions for Authors

The integration of hybrid fruits into dietary practices has emerged as a compelling avenue in the pursuit of enhanced health and nutrition. This comprehensive review endeavors to consolidate and analyze the burgeoning body of knowledge surrounding the potential health benefits, nutritional value, and safety considerations associated with the consumption of hybrid fruits. By amalgamating insights from diverse scientific studies, this paper aims to elucidate the diverse spectrum of hybrid fruits, exploring their composition, bioactive components, and their purported contributions to holistic well-being. Furthermore, this review delves into the regulatory frameworks and safety evaluations governing these novel fruit hybrids, shedding light on their acceptance within the purview of governmental agencies and scientific communities. Through an extensive survey of literature and critical synthesis, this review endeavors to navigate the landscape of hybrid fruits, offering a nuanced understanding of their role in fostering improved health outcomes and dietary diversity.

The authors should try to improve the paper's quality by following the next suggestions:

The authors made changes to the article to improve its quality, following the suggestions. The changes are presented in red letters in the attached article.

Abstract:

The study does not offer a demonstration of findings as it is not a research paper. It effectively provides an overview and synthesis of existing information on hybrid fruits as an alternative for health improvement.

We include in the abstract, the information that the study provides an overview and synthesis of existing information on hybrid fruits for the improvement of health.

Introduction (Lines 61-64):

The mentioned approach seems to be aligned with the principles outlined in the provided paper (https://doi.org/10.3390/foods12214001). Incorporating similar methodologies could enhance the clarity and depth of this study's approach.

We have included similar methodologies (lines 61 to 67), in red letters, to improve the clarity and depth of the approach of this study.

Introduction (Repetitive References):

The introductory section seems to have certain references reiterated, potentially elevating the similarity index of the paper. Diversifying the sources cited could enhance the comprehensiveness of the introductory part.

Modifications were made to the introduction to increase the scope of the introductory part, as suggested. Changes are marked in yellow and/or red letters.

Line 100-104:

Consider revising this segment for greater clarity as the intended message might not be distinctly conveyed. A clearer articulation of the concept would enhance the reader's understanding.

The segment was revised and changed through the articulation of the concept, for greater understanding of the reader.

Amendment tabled in red letters, lines 96 to 102.

Figure 1:

While the visual presentation in Figure 1 is commendable, the size renders it difficult to read. Enlarging the figure would significantly improve its accessibility and utility for readers.

Figure 1 has been improved, as suggested, to increase the number and significantly improve its accessibility and usefulness to readers.

FDA Mention (Line 140):

For the first instance, it's advisable to spell out the abbreviation "FDA" as "Food and Drug Administration" for clarity before using the acronym.

The abbreviation "FDA" was written as "Food and Drug Administration" for clarity before using the acronym (152-153).

References in Section 2.3 and Lines 139-163:

Consider adding pertinent references to substantiate the claims made in these sections. Strengthening these assertions with reliable sources would fortify the paper's credibility.

We have included the references “[17-20]” to substantiate the claims made in these sections.

Section 2.3:

The structure and clarity of Section 2.3 could benefit from improvement. Enhancing the organization and coherence of this section would enhance its readability and comprehension.

The structure and clarity of section 2.3 have been revised to improve readability and comprehension.

Tables 9 and 10:

Tables 9 and 10, currently presented as images, require better clarity and accessibility. Transforming them into readable formats and discussing the data within these tables would enhance their informative value.

Tables have been transformed into readable formats.

Proper Citation (Line 629):

Ensure proper citation of references for accuracy and academic integrity.

Citations of the references were checked.

Visuals:

The colorful and logical presentation of images is highly commendable. They effectively complement the content.

 Thank you very much for your observation.

In conclusion, enhancing the logical sequencing of content, providing more comprehensive data, specifying the discussed hybrid fruits, and reinforcing claims with relevant references would substantially augment the quality and impact of this paper.

We seek to improve the quality and impact of the article by using the suggestions provided and greatly appreciate the opportunity to present the resource. We wish we had complied with the richly oriented suggestions.

Reviewer 2 Report

Comments and Suggestions for Authors

The review by Cruz et al. discusses hybrid fruits as an alternative for improving health. They suggest that hybrid fruits are a source of antioxidant compounds, and bioactive compounds They also suggest a possible  role of these substances in the inhibition of diseases such as cancer, diabetes, inflammatory and neurodegenerative diseases. The manuscript is well-written and well-organized. However, some points should be addressed before publication.

- Title: The Authors should change the word alternative. Writing alternative is underestimating.

- More studies supporting the idea that a diet rich of fruits is effective for improving health must be added and discussed. For instance, the following studies (DOI: 10.3390/ijms21228653; DOI:10.3233/JBR-220054 and others) supporting the concept that diets including fruits rich in anthocyanins can improve in general mental health (not only neurodegenerative disorders) must be added and discussed. More studies about  fruits and neuropsychiatric symptoms must be included.

- The discussion about potential mechanisms explaining the beneficial effects of hybrid fruits should be enlarged. This would improve the quality of the manuscripts.

-There are several typos throughout the manuscript.

Comments on the Quality of English Language

Moderate editing is required

Author Response

Anderson Junger Teodoro, PhD

Department of Nutrition and Dietetics

Universidade Federal Fluminense

email: atteodoro@gmail.com

Editor

Food Journal

Foods(https://www.mdpi.com/journal/foods)

Dear Editor,

Find an electronic version of the revised manuscript entitled Hybrid fruits as an alternative for improving health - a comprehensive review. (Submission Ref. Manuscript ID: foods-2784565).

We would like to thank the reviewers for all their comments and suggestions that helped us improve our work and present it in a more acceptable way. The response to the comments raised by the reviewers, as well as the list of changes made to the revised manuscript, are attached below, in red.

All changes made to the manuscript have been highlighted as requested. We hope that this revised version complies with the Reviewers' suggestions and is suitable for publication in his journal.

Best regards,

Anderson Junger Teodoro, PhD

Highlights comments:

The version of the manuscript found in the link above was used for the Reviews.

https://susy.mdpi.com/user/manuscripts/resubmit/c76bd2da5afbba04ed9c4122ac558cdc

(I) All references are relevant to the content of the manuscript.

(II)  The revisions of the manuscript were highlighted, so that any of changes can be easily reviewed by editors and reviewers.

(III) This cover letter explains, point by point, the details revisions of the manuscript and their responses to the reviewers Comments.

(IV) All comments in the review are explained in the resource.

(V) The revised version is attached to the editors and reviewers.

Reviewer #2:

comments and Suggestions for Authors

The review by Cruz et al. discusses hybrid fruits as an alternative for improving health. They suggest that hybrid fruits are a source of antioxidant compounds, and bioactive compounds They also suggest a possible role of these substances in the inhibition of diseases such as cancer, diabetes, inflammatory and neurodegenerative diseases. The manuscript is well-written and well-organized. However, some points should be addressed before publication.

We appreciate the suggestions and enriching points suggested to our review.

- Title: The Authors should change the word alternative. Writing alternative is underestimating.

The title has been changed as suggested, to better understand the authors' expectations.

=>   Hybrid fruits for improving health - a comprehensive review

- More studies supporting the idea that a diet rich of fruits is effective for improving health must be added and discussed. For instance, the following studies (DOI: 10.3390/ijms21228653; DOI:10.3233/JBR-220054 and others) supporting the concept that diets including fruits rich in anthocyanins can improve in general mental health (not only neurodegenerative disorders) must be added and discussed. More studies about fruits and neuropsychiatric symptoms must be included.

More references were included, as suggested.

- The discussion about potential mechanisms explaining the beneficial effects of hybrid fruits should be enlarged. This would improve the quality of the manuscripts.

The rewriting of some paragraphs and the inclusion of new references, in red letters, were carried out to meet the suggestion.

-There are several typos throughout the manuscript.

We performed a review of typos throughout the manuscript.

Round 2

Reviewer 1 Report

Comments and Suggestions for Authors

Reference no 7 I think is wrong. Please check 

Reviewer 2 Report

Comments and Suggestions for Authors

The Authors have ameliorated the work.

Comments on the Quality of English Language

Minor editing